# Age-Friendly Environments in ASEAN Plus Three: Case Studies from Japan, Malaysia, Myanmar, Vietnam, and Thailand

**DOI:** 10.3390/ijerph17124523

**Published:** 2020-06-23

**Authors:** Sariyamon Tiraphat, Doungjai Buntup, Murallitharan Munisamy, Thang Huu Nguyen, Motoyuki Yuasa, Myo Nyein Aung, Aung Hpone Myint

**Affiliations:** 1ASEAN Institute for Health Development, Mahidol University, Salaya, Phutthamonthon, Nakhon Pathom 73170, Thailand; doungjai.bun@mahidol.ac.th; 2National Cancer Society of Malaysia, Kuala Lumpur 50300, Malaysia; murallimd@gmail.com; 3Institute for Preventive Medicine and Public Health, Hanoi Medical University, Hanoi 100000, Vietnam; nguyenhuuthang@hmu.edu.vn; 4Faculty of International Liberal Arts, Juntendo University, Tokyo 113-8421, Japan; moyuasa@juntendo.ac.jp (M.Y.); dr.myonyeinaung@gmail.com (M.N.A.); 5Community Partners International (CPI), Bahan Township, Yangon 11201, Myanmar; aunghponemyint88@gmail.com

**Keywords:** age-friendly environments, emergency response, lifelong learning, political, economic, health, personal care, village health volunteer, ASEAN plus three, older populations

## Abstract

Promoting age-friendly environment is one of the appropriate approaches to support quality of life toward ageing populations. However, the information regarding age-friendly environments in the Association of Southeast Asian Nations (ASEAN) Plus Three countries is still limited. This study aimed to survey the perceived age-friendly environments among ASEAN Plus Three older populations. The study employed cross-sectional quantitative research using multistage cluster sampling to select a sample of older adults in the capital cities of Japan, Malaysia, Myanmar, Vietnam and Thailand. The final sample was composed of 2171 older adults aged 55 years and over, including 140 Japanese, 510 Thai, 537 Malaysian, 487 Myanmarese, and 497 Vietnamese older adults. Data collection was conducted using a quantitative questionnaire with 20 items of perceived age-friendly environments with the rating scale based on the World Health Organization (WHO) standard. The score from the 20 items were analyzed and examined high-risk groups of “bad perception level” age-friendly environments using ordinal logistic regression. The research indicated the five highest inadequacies of age-friendly environments including: (1) participating in an emergency-response training session or drill which addressed the needs of older residents; (2) enrolling in any form of education or training, either formal or non-formal in any subject; (3) having opportunities for paid employment; (4) involvement in decision making about important political, economic and social issues in the community; and (5) having personal care or assistance needs met in the older adult’s home setting by government/private care services. Information regarding the inadequacy of age-friendliness by region was evidenced to guide policy makers in providing the right interventions towards older adults’ needs.

## 1. Introduction

Globally older populations are increasing more than other age groups with a faster rate in the developing countries [1]. Information from the United Nations indicated that in the year 1980 the older population was a majority (56% of persons aged 60 years or over) in developed countries. However, accordingly, in the future (2050) almost 80% of the older population aged 60 and over will be found in the less-developed countries [1]. Throughout the world, ageing populations have been the fastest growing in East Asia and Southeastern Asia where the percentage of the population aged 65 years or over almost doubled from 6% in 1990 to 11% in 2019 [2].

Recognizing the growing trend of a worldwide ageing society, the World Health Organization (WHO) introduced the concept of an “age-friendly community” in 2005 and published a guide in 2007 [3]. There has been rapid growing interest in making communities more age-friendly and a large number of research studies have been undertaken on age-friendly environments based on the WHO guideline. For example, recent studies from Europe implemented the concept of age-friendly cities in The Netherlands and Poland. The studies illustrate the potential of making cities more appropriate to the needs of older people and to identify important challenges for active ageing in current and future generations. Some challenges involve the establishment of inclusive neighborhoods, such as making accessible neighborhoods with an adequate provision of services as well as the implementation of technology, such as using smart home-monitoring technologies for ageing societies [4,5,6,7]. In Canada, some researchers developed age-friendly indicators. The final list includes 39 indicators across eight domains that can support communities in their evaluation activities [8]. The domains are: outdoor space and buildings, transportation, housing, social participation, respect and social inclusion, civic participation and employment, communication, and community support and health services. Another study from Canada recommended the designing of age-friendliness for all states by promoting strategic engagements such as strengthening collaborative intersectoral relationships, implementing policy actions such as funding community projects, and development and exchange of knowledge such as the creation of a research community and policy network [9]. Several previous researchers in the United States also studied developing age-friendliness towards the older population such as improving the workforce in the field of aging [10,11], strengthening social capital such as social connectivity associated with the older population’s health benefits [12,13], and integrating research into a policy and planning agenda such as offering a policy to improve the physical and social environments for seniors [14,15,16]. In addition to Europe and America, many Australian researchers conducted studies regarding age-friendliness to promote quality of life towards Australian adults such as assessing the impact of political, physical, social, and research dimensions to implement ‘ageing in place’ in Australia [17,18,19,20].

In Asia, the Association of Southeast Asian Nations (ASEAN) Plus Three (APT) cooperation was founded to strengthen and deepen East Asia cooperation in various areas, particularly in economic, social, and political fields. Currently, APT includes the 10 members of the Association of Southeast Asian Nations (Brunei, Cambodia, Indonesia, Laos, Malaysia, Myanmar, the Philippines, Singapore, Thailand, and Vietnam) plus China, Japan, and South Korea. In 2016, to address and prepare for ageing societies, the APT presented a “statement on active ageing” in a purposeful manner to support the quality of life and well-being of the older persons in the regions [21]. To promote active ageing of older residents, a holistic approach of supportive and “age-friendly environments” for the older population included not only social and physical environments, but also economic security, and health care services as a national priority [21,22]. 

Despite the fact that ageing populations have been growing rapidly in the ASEAN Plus Three, a study to investigate age-friendly environments for older adults in the region is limited. Strengthening timely and effective policy cooperation towards active ageing, information and evidence regarding overview of age-friendly environments would be necessary. Therefore, this study aimed (1) to examine the availability of provided age-friendly environments; (2) to investigate whether the distribution of provided age-friendly environments differ by country; and (3) to identify high-risk groups of having inadequate age-friendly environments. Evidence of perceived age-friendly environment and risk groups will envision health and social authorities to create appropriate age-friendly environments based on the older populations’ needs.

## 2. Methodology

### 2.1. Description of Survey and Study Population

This research design was a cross-sectional household interview survey of perceived age-friendly environments towards the older populations in ASEAN Plus Three. The study used a multistage, stratified sampling procedure collecting data via face-to-face interviews during November 2018 to January 2019 in five metropolitan areas of Malaysia, Myanmar, Vietnam, Thailand, and Japan. We calculated the sample size using the number of older population aged 55 years and older in each metropolitan area above (100,000 cases/metropolis) [2,23]. Having been calculated by the Taro Yamane Formula method with 95% confidence, the sample size in each metropolitan area was about 400 cases. After adding a missing rate of 25%, the expected final sample size for each metropolitan area was approximately 500 cases. However, we decided to have smaller numbers of data collection (about 150 cases) in Japan due to our resource limitations. For all countries, the first step was selection of a metropolitan area. In the second step, we randomly selected three or four districts from the metropolitan areas. In the third step, each subdistrict per district was selected. In the fourth step, every person 55 years of age and older living in the randomly selected households in the study area was eligible for the study. In the final fifth step, among all the eligible respondents in a household, one was randomly selected for interview. The response rate in each country was 100%. The study population after excluding the observations with missing data was a total of 2171 persons aged 55 years and older. The observations remaining in the sample included 537 from Malaysia, 497 from Vietnam, 487 from Myanmar, 510 from Thailand, and 140 from Japan. This research project received ethical approval from the “Research Ethics Committee of the Faculty of Social Sciences and Humanities, Mahidol University” (Certificate of Approval No. 2018/218.1809). Informed consent was obtained from all study participants.

### 2.2. Measures

#### 2.2.1. Perceived Age-Friendly Environments

All items of perceived age-friendly environments scale were obtained from the World Health Organization [3,24]. The final questionnaire of this study was adapted from the age-friendly environment questionnaire’s Thai version, which has been validated with an older adult population in Thailand showing a good internal consistency validity α = 0.89 [25]. The perceived age-friendly environment scale is composed of 20 items for evaluating 8 domains that cities and communities can address to better adapt their structures and services to the needs of older people. The domains are the built environment, transport, housing, social participation, respect and social inclusion, civic participation and employment, communication, and community support and health services [3]. Each individual item of the age-friendly environments is scored from 0 to 4 on a response ordinal scale (not at all, a little, moderately, mostly, extremely), with higher scores indicating a higher perception of age-friendly environments. In this study, to examine whether distribution of provided age-friendly environments differ by country, we integrated and classified the perceived age-friendly environments from five levels into three levels as bad (not at all/a little), fair (moderately), and good (mostly/extremely). Internal consistency for the perceived Age-friendly environments in this study sample was α = 0.87, with 0.80, 0.67, 0.79, 0.88, and 0.84 for Malaysia, Vietnam, Myanmar, Thailand, and Japan respectively.

#### 2.2.2. Sociodemographic Variables

In order to identify high-risk groups of perceived age-friendliness inadequacy, we put age levels as 1 = 55–64 years, 2 = 65–74 years, 3 = 75 years and higher, and designated gender, and placed educational levels at 1 = At least primary school, 2 = High school, and 3 = More than high school as the predictors.

### 2.3. Data Analysis

Data were analyzed using the 2012 released IBM SPSS Statistics for Windows, Version 21.0, (Armonk, NY, USA). Descriptive analysis was used to describe the sample. In order to survey the availability of perceived age-friendly environment, the average score of each individual item of the response ordinal scale (0–4) for age-friendly environments was calculated with the average higher scores indicating more availability of age-friendly environments. In order to examine differences in the proportion of perceived age-friendly environment by country, we recoded the perception into 3 levels as bad (not at all/a little), fair (moderately), and good (mostly/extremely) and did the analysis using Pearson chi-squared test. Finally, to identify high-risk groups experiencing inadequate age-friendliness, ordinal logistic regression analysis was applied to investigate the predictors including age level, gender, educational level, and participant’s country of higher perceived age-friendly environment. The level of significance for all analyses was set at *p* < 0.05. 

## 3. Results

### 3.1. Sample Characteristics

The total study sample included 2171 older persons (55 years or more). About three-fifths (61.6%) of the sample were women. They had completed elementary school (42.1%), high school (24.3%), and more than high school (33.7%). The older adults are more in the ages between 55–64 years (44.6%), followed by 38.2% for the ages between 65–74 years, and 17.2% for the older population at 75 years or higher. They live in Malaysia, Vietnam, Myanmar, Thailand, and Japan with *N* = 537, 497, 487, 510, and 140 respectively (see Table 1). 

### 3.2. Availability of Perceived Age-Friendly Environments

Availability of perceived age-friendly environments are analyzed from the scores for each one of the 20 items of an age-friendly environment. Average score of each individual item of the response ordinal scale (0–4) for age-friendly environments was calculated and compared with average higher scores indicating more availability of age-friendly environments.

Table 2 indicates average score of perceived age-friendly environments. We found that the five lowest average scores of perceived age-friendly environment in ASEAN Plus Three are as follows: (1) participating in an emergency response training session or drill in the past year which addressed the needs of older residents (mean score = 0.58); (2) enrolling in any form of education or training, either formal or non-formal, in any subject in the past year(mean score = 1.04); (3) having opportunities for paid employment (mean score = 1.06); (4) involving in decision making about important political, economic and social issues in the community(mean score = 1.07); and (5) having personal care or assistance needs met in a home setting, e.g., home care nursing/hospice care/non-governmental organization (NGO)/volunteers (mean score = 1.10).

In contrast, the five highest average score of perceived age-friendly environments in ASEAN Plus Three are as follows: (1) feeling safe in the neighborhood (mean score = 2.61); (2) feeling respected and socially included in the community (mean score = 2.18); (3) local sources of information about your health concerns and service needs are available (mean score = 1.91); (4) the neighborhood is suitable for walking, including for those who use wheelchairs and other mobility aids (mean score = 1.89); and (5) house has been renovated, or can be renovated to fulfil needs in order to support the activities of daily living (mean score = 1.86).

### 3.3. The Level of Perceived Age-Friendly Environments by Country

In order to examine the differences in the proportion of perceived age-friendly environments by country, we recoded the perception into 3 levels as bad (not at all/a little), fair (moderately), and good (mostly/extremely). A Pearson chi-squared test was applied to examine the difference in the level of perceived age-friendly environments by country. The results from chi-squared test in Table 3, indicated the significant differences at *p* < 0.05 in the proportion of all perceived age-friendly environments by country. Among the 20 items of age-friendly environments, the five highest percentage of inadequate age-friendly environments are (1) participating in an emergency response training session or drill in the past year which addressed the needs of older residents; (2) enrolling in any form of education or training, either formal or non-formal; (3) having opportunities for paid employment; (4) involving in decision making about important political, economic and social issues in the community; and (5) having the personal care or assistance needs met in home setting by government/private care services. Namely, most ASEAN older population (more than 80%) perceived their emergency-response training session or drill as bad, especially more than 90% of the Vietnamese and Myanmarese older populations. Regarding education or training, almost 70% of the older adults perceived badly, especially most Myanmarese older adults (95.9%). Similar to education or training, paid employment is also a big problem for the older population (almost 70%) with inadequate opportunity to get employment, especially Myanmarese older adults (96.9%). Regarding decision making about important political, economic and social issues in the community, 65% of the older population perceived it badly, especially 97.1% of Myanmarese and 77.1% of Malaysian older adults are not satisfied with it. For personal care or assistance needs met in their home setting, almost 65% of the older population perceived this to be inadequate, especially 96.7% of Myanmarese and 93.2% of Vietnamese older adults are not satisfied; however the result showed that only 29% of Thai older adults are dissatisfied.

### 3.4. High-Risk Group of Having Inadequate Age-Friendly Environments 

We identify a high-risk group of having inadequate age-friendly environments by analysis of ordinal logistic regression with *p* < 0.05. The interested predictors include age level, gender, educational level, and participant’s country. From Table 4, regarding *Model 1**, the high-risk group of perceived low level of participating in an emergency response training session or drill which addressed the needs of older residents is the older population with low educational level. Namely, the results indicated that older adults with more than high school are significantly more likely to rate for a higher scale of perceived emergency-response training sessions or drills compared to the older adults with at least primary school, controlling for other socioeconomic statue (SES) and country of participants.

Regarding *Model 2**, high-risk groups of perceived low level of education or training enrollment are the older population with low educational level and the oldest group. Apparently, the results indicated that older adults with the lowest educational level are significantly less likely to rate for a higher scale of perceived education or training enrollment compare to the older adults with the highest educational level, controlling for other SES and country of participants. Additionally, the older adults with 75 years and higher are significantly less likely to rate for a higher scale of perceived education or training enrollment compare to the older adults aged 55–64 years, controlling for other SES and country of participants.

Regarding *Model 3**, high-risk groups of perceived low opportunities for paid employment are the older population with low educational level and the oldest group. Obviously, the results indicated that older adults with lower educational level are significantly less likely to rate for a higher scale of perceived opportunities for paid employment compared to the older adults with the highest educational level, controlling for other SES and country of participants. Additionally, the older adults aged 75 years and higher are significantly less likely to rate for a higher scale of perceived opportunities for paid employment compared to the older adults that are younger, controlling for other SES and country of participants.

Regarding *Model 4**, high-risk groups of perceived low involvement in decision making about important issues in the community are the older adult population with low educational level and the female older adult population. Especially, the results indicated that older adults with lower educational level are significantly less likely to rate for a higher scale of perceived involvement in decision making about important political, economic and social issues in the community compared to the older adults with highest educational level, controlling for other SES and country of participants. Additionally, female older adults are significantly less likely to rate for a higher scale of perceived involvement in decision making about important political, economic and social issues in the community compared to male older adults, controlling for other SES and country of participants.

Regarding *Model 5**, the high-risk group of perceived low level of having personal care or assistance needs was the male older adult population. Evidently, the results indicated that male older adults are significantly less likely to rate for a higher scale of perceiving having personal care or assistance needs met compared to the female older adults, controlling for other SES and country of participants.

## 4. Discussion

The study found significant differences in the proportion of perceived age-friendly environments by ASEAN Plus Three older populations. The analysis results from the chi-squared test and ordinal logistic regression identified that among the 20 items of age-friendly environments, the five highest unsatisfied age-friendly environments toward ASEAN Plus Three ageing population are: (1) the inadequacy of participating in an emergency response training session or drill in the past year which addressed the needs of older residents, especially in the older population with low educational level; (2) the inadequacy of enrolling in any form of education or training, either formal or non-formal, especially in the older population with low educational level and the oldest group; (3) the inadequacy of having opportunities for paid employment, especially in the older population with low educational level and the oldest group; (4) the inadequacy of involving in decision making about important political, economic and social issues in the community, especially in the older population with low educational level and female older adults; and (5) the inadequacy of having the personal care or assistance needs met in home setting by government/private care services, especially in male older adults.

There are some similarly satisfied age-friendly environments toward the ageing population in ASEAN Plus Three. Namely, the older population in Myanmar, Vietnam, and Japan similarly perceived a score of “they feel safe in their neighborhood” with the highest of all environments, whereas the Thai and Malaysian older population also perceived this item with the third highest of all. The most satisfied environments “feeling safe in their neighborhood” and “feeling respected in the community” was especially liked by Thai, Vietnamese, and Myanmarese older populations.

For the high-risk groups, the statistical analysis of ordinal logistic regression indicated that the older population with lower level of education is the high-risk group of “bad” perceived environment, especially for training issues, paid employment, and making decisions about important issues. Additionally, the study indicated that oldest aged adults was the high-risk group that significantly perceived a “bad” environment regarding education or training enrollment, and paid employment. Finally, we found gender disparity in the perception of decision making about important issues of males to be superior, and having personal care or assistance needs of females to be superior.

In agreement with previous researchers [26,27], our results indicated that ASEAN Plus Three older populations feel safe in their neighborhood. Past research also showed very high levels of trust and co-operation in the neighborhood among residents of Southeast Asia and higher levels of contribution towards the residents than those in North America and Europe [26]. A previous study [27] claimed that “increasing physical and cognitive constraints from being old may move them to gain more difficulties in completing some challenging tasks by themselves. Therefore, having more trust toward others contributes to the older adults more comfortableness to accept and rely on others’ help”. Additionally, it is noted that most people in ASEAN countries regularly live in the same place for generations and root deeply in the neighborhood. For these reasons, it is reasonable to identify that the older adults may have high levels of trust toward their neighbors [27]. In addition to perceived neighborhood safety, the results of our study also strongly support the respect of older adults of ASEAN generations. In the same line with our results, previous researchers indicated that respect for older adults is the most stressed expression of filial piety and it is deeply rooted in traditional Asian cultures [28,29,30]. The value of respect has retained its stability in the region for generations, however, evidence of changing respect expressions [30] including gestures and manners, tokens, customs and rituals, asking for advice, and obedience has occurred. Main factors associated with changing may include variations in family structure and function, education, income, and modernization [29]. Thus, it is a big challenge how ASEAN populations will retain deeply rooted values in a changing world.

Interestingly, this is the first study to indicate the inadequacy of emergency-response training toward older populations in ASEAN Plus Three, especially the older population with low education. In fact, aging brings many disadvantages to the older adults due to their physical, mental, and cognitive impairment-related aging process. In recent years [31] assistive technologies, such as the mobile and wearable sensors, assistive robots, smart homes, and smart fabrics for emergency response were introduced to maintain the independence of older populations, as well as to monitor and improve their health conditions. Although emergency assistive technologies are useful for older adults, previous research indicated [32] that aged populations even in a modern country such as Japan, have a more negative attitude towards performing basic life support. Therefore, there are still many challenges to help the older adults gain more confidence and skill with the essential elements of emergency response, especially those with low educational level. Besides emergency training, our results indicated the inadequacy of enrolling in any form of education or training, either formal or non-formal, or lifelong learning, especially in the older adults with low education and the oldest group. There is evidence that lifelong learning could promote older adults’ health and well-being [33,34]. Therefore, some researchers tried to investigate appropriate practical courses for older adults and found that languages and health-related topics were the most popular among the older adults, especially in China [35] and the USA [36]. Interestingly, our study also showed that Japanese older adults living in modern society perceive an inadequate quantity of education and training to be supplied. Living in a nation with high literacy and technically advances in science and technology may be the reason why Japanese older adults are being challenged to constantly acquire new knowledge and skills [37]. The evidence of this study indicates a challenge for educators to provide continuing education opportunities with various and appropriate practices towards the older adults in ASEAN Plus Three countries.

Regarding the perceived paid employment, it is evident that almost 70% of older adults are in need of paid employment support, especially the older populations with less education. Along the same lines as our results, research showed that older adults in less developed countries are more likely to face economic necessities, especially the uneducated workers [38]. Our results also found that even the older populations in Japan are in need of employment. Consistent with the results of our study, a previous study [38] indicated that Japanese older adults prefer to extend their work after retirement. The researcher explained that the reasons that Japanese older adults prefer to keep working are: (1) they want to keep their standard of living as it was in their late 50s; and (2) they are concerned about society’s norms that value the older adults staying in the labor force as long as possible. Some other countries in ASEAN, such as Thailand and Vietnam, attempt to initiate a national plan to delay the retirement age for maintaining the older adults at work, thus leading to more active ageing and economic security. Our results indicate the need of work at retirement age for all regions and, therefore, it is a challenge to policy makers how to allocate appropriate work for the older populations with long-term experiences but minor difficulty in physical conditions.

Decision-making about important political, economic and social issues in the community is another inadequate environment item of the older populations, especially the older adults with low education and being female. Like our research, a previous study found disparity of social participation including collective, productive, and political participation by socioeconomic status [39]. The researcher addressed that “older men are more likely to be engaged in paid work outside the home, even after retirement, in political activities and clubs, whereas older women more often take care of children (or grandchildren) and do more volunteer work and caregiving outside the home”. Another study also supported that [40] older adult males have an important role in making decisions on important issues such as economic or political subjects, whereas older adult women tend to provide non-economic contributions to families, such as women’s health or social services volunteers in communities. In addition to gender, research [39] also stressed that “persons who possess more educational and occupational resources may participate in social participation longer than persons with fewer resources, even after their health declines”.

Lastly, an unsatisfied age-friendly environment is about inadequacy of the personal care or assistance needs met in a home setting, especially in Vietnam and Myanmar. Compared with the older adults in other countries, our results confirmed that older adults in Thailand have most satisfaction with the personal care or assistance needs at home. It is evident that all communities throughout the country of Thailand are equipped with strong community-based care for the older adult populations with more than 20 years countrywide ‘elders’ clubs’ offered for older adult people [41]. Moreover, the country exists with the village health volunteers (VHVs) handling the older adults at home that have long been recognized by the World Health Organization as an international model for community-based public health. These reasons may explain why Thai older adult populations are more satisfied with personal care or assistance needs met in their home [42].

The results of this study can assist policy planners in building more appropriate age-friendly environments towards older adult populations in the ASEAN Plus Three. The priorities of environments for aging adult populations should be boosted towards active ageing as follows: (1) increasing an emergency-response training session or drill which addresses the needs of older residents, especially older adults with lower education; (2) giving any form of education or training, either formal or non-formal especially to older adults with lower education and the oldest group; (3) maintaining opportunities for paid employment for older adults in need; (4) supporting social participation and decision making about important political, economic and social issues in the community, especially for female and low-educated older adults; and (5) increasing the personal care or assistance needs met in a home setting by government/private care services (e.g., home care nursing/hospice care/non-governmental organization (NGO)/volunteers)”, especially in male older adult populations.

There are some limitations to this study. First, the nature of the cross-sectional design of the study cannot confirm the causal relationships between age-friendly environments and the predictors. Second, data collection is diversified as the trained researchers in each country may have different skills, and thus, the study may have data collection bias. Third, the measurement of age-friendly environments in this study relied on perceived rather than objective measures of the environments. However, the present study has its strengths as it is the first study to survey age-friendly environments in ASEAN Plus Three, therefore, gaining knowledge and evidence for societies being cooperative in catering for ageing adult populations in these regions.

## 5. Conclusions

The growing trend of worldwide ageing adult populations is the main challenge of creating age-friendly environments. The present study significantly indicates the perception of the older adult populations’ concerns about age-friendly environments in the ASEAN Plus Three countries. In order to help communities to become age-friendly, priorities of environmental improvement need to be considered including: (1) increasing emergency-response training sessions or drills that addresses the needs of older adult residents: (2) giving any form of education or training, either formal or non-formal for older adults: (3) maintaining opportunities for paid employment for the older adults in need; (4) supporting social participation and decision making about important political, economic and social issues in the communities; and (5) increasing the personal care or assistance needs met in home settings by government/private care services.

## Figures and Tables

**Table 1 ijerph-17-04523-t001:** Sample characteristics.

Variables	Country											
	Malaysia (*N* = 537)		Vietnam (*N* = 497)		Myanmar (*N* = 487)		Thailand (*N* = 510)		Japan (*N* = 140)		Total (*N* = 2171)	
	*N*	%	*n*	%	*N*	%	*n*	%	*N*	%	*n*	%
1. Education												
1.1 At least Primary school	19	3.5	165	33.2	415	85.2	313	61.4	1	0.7	913	42.1
1.2 High school	90	16.8	247	49.7	68	14.0	110	21.6	12	8.6	527	24.3
1.3 More than high school	428	79.7	85	17.1	4	0.8	87	17.1	127	90.7	731	33.7
2. Gender												
2.1 Male	233	43.4	212	42.7	164	33.7	146	28.6	78	55.7	833	38.4
2.2 Female	304	56.6	285	57.3	323	66.3	364	71.4	62	44.3	1338	61.6
3. Age level												
3.1 55–64 years	376	70.0	185	37.2	201	41.3	200	39.2	6	4.3	968	44.6
3.2 65–74 years	144	26.8	218	43.9	191	39.2	194	38.0	83	59.3	830	38.2
3.3 75 years and higher	17	3.2	94	18.9	95	19.5	116	22.7	51	36.4	373	17.2

**Table 2 ijerph-17-04523-t002:** Average score of perceived age-friendly environments calculated from the average of a five rating scale (0–4) with 0 = not at all, 1 = a little, 2 = moderate, 3 = mostly, 4 = extremely.

Items of Age-Friendly Environment	Average Score of Perceived Age-Friendly Environments by Country					
	Malaysia (*N* = 537)	Vietnam (*N* = 497)	Myanmar (*N* = 487)	Thailand (*N* = 510)	Japan (*N* = 140)	Total (*N* = 2171)
1. Your neighborhood is suitable for walking, including for those who use wheelchairs and other mobility aids.	1.67 (SD = 1.08)	2.63^highest5^ (SD = 0.93)	1.60^highest2^ (SD = 0.91)	1.74 (SD = 1.09)	1.71 (SD = 0.98)	1.89^highest4^ (SD = 1.09)
2. The public spaces and buildings in your community are accessible for all people, including those who have limitations in mobility, vision or hearing.	1.45 (SD = 1.10)	2.48 (SD = 0.97)	0.55 (SD = 0.92)	1.65 (SD = 1.12)	1.49 (SD = 0.82)	1.53 (SD = 1.21)
3. The public transport vehicles (e.g., train cars, buses) are physically accessible for all people, including those who have limitations in mobility, vision or hearing.	1.19^lowest 3^ (SD = 1.09)	2.25 (SD = 1.11)	0.75 (SD = 1.10)	1.38 (SD = 1.16)	1.49 (SD = 0.93)	1.40 (SD = 1.22)
4. The public transportation stops (such as bus stops) are not too far from your home.	1.49 (SD = 1.04)	2.35 (SD = 1.14)	0.31 (SD = 0.72)	1.48 (SD = 1.13)	1.92 (SD = 1.14)	1.45 (SD = 1.25)
5. Housing in your neighborhood is affordable.	1.49 (SD = 0.83)	2.75^highest4^ (SD = 0.86)	0.02^lowest1^ (SD = 0.21)	1.86^highest5^ (SD = 0.97)	1.84 (SD = 0.88)	1.56 (SD = 1.22)
6. You feel respected and socially included in your community.	2.21*^highest4^* (SD = 0.83)	3.08^highest2^ (SD = 0.81)	1.38^highest3^ (SD = 1.29)	2.22^highest1^ (SD = 0.73)	1.53 (SD = 0.88)	2.18^highest2^ (SD = 1.11)
7. Your neighborhood provide volunteer activity to the older in the last month on at least one occasion.	1.74 (SD = 1.25)	0.72^lowest3^ (SD = 1.07)	0.72 (SD = 1.13)	1.68 (SD = 1.07)	1.14^lowest3^ (SD = 1.35)	1.23 (SD = 1.25)
8. You have opportunities for paid employment (i.e., there are opportunities for you to get a paid job if you want for an older person).	1.53 (SD = 1.12)	1.39 (SD = 1.36)	0.12^lowest3^ (SD = 0.50)	1.20 (SD = 1.06)	0.88^lowest1^ (SD = 1.33)	1.06^lowest3^ (SD = 1.21)
9. Your neighborhood provided sociocultural activities to the older at least once in the last week.	1.64 (SD = 1.13)	0.93^lowest4^ (SD = 1.22)	0.69 (SD = 1.10)	1.89^highest4^ (SD = 0.98)	1.40 (SD = 1.32)	1.31 (SD = 1.22)
10. You are involved in decision making about important political, economic and social issues in the community.	0.74^lowest2^ (SD = 0.93)	1.88 (SD = 1.26)	0.14^lowest4^ (SD = 0.61)	1.40 (SD = 1.11)	1.46 (SD = 1.31)	1.07^lowest4^ (SD = 1.21)
11. Local sources of information about your health concerns and service needs are available.	2.12*^highest5^* (SD = 1.23)	2.35 (SD = 1.05)	1.05*^highest5^* (SD = 1.08)	2.03*^highest3^* (SD = 1.06)	2.12*^highest3^* (SD = 1.06)	1.91*^highest3^* (SD = 1.21)
12. You have your personal care or assistance needs met in your home setting by government/private care services (i.e., home care nursing/hospice care/non-governmental organization (NGO)/volunteers).	1.31*^lowest4^* (SD = 1.14)	0.44*^lowest1^* (SD = 0.80)	0.34 (SD = 0.63)	2.05*^highest2^* (SD = 1.18)	1.83 (SD = 0.96)	1.10*^lowest5^* (SD = 1.19)
13. You have had enough income to meet your basic needs over the previous 12 months without public or private assistance.	2.7*^highest2^* (SD = 0.96)	2.32 (SD = 1.36)	0.76 (SD = 0.89)	1.57 (SD = 0.97)	1.51 (SD = 1.15)	1.85 (SD = 1.29)
14. Designated priority parking spaces are adequately designed and available.	1.46 (SD = 1.15)	2.46 (SD = 1.04)	0.17 (SD = 0.55)	1.12*^lowest2^* (SD = 1.13)	2.19*^highest2^* (SD = 1.01)	1.37 (SD = 1.29)
15. Your house has been renovated, or can be renovated to fulfil your needs in order to support your activities of daily living.	2.06 (SD = 1.09)	2.81*^highest3^* (SD = 0.99)	1.16*^highest4^* (SD = 1.11)	1.37 (SD = 1.13)	1.95*^highest5^* (SD = 1.09)	1.86*^highest5^* (SD = 1.25)
16. Your neighborhood provided group physical activities in your leisure time.	1.64 (*SD* = 1.10)	1.52 (*SD* = 1.44)	0.25 (*SD* = 0.70)	1.37 (*SD* = 1.10)	1.69 (*SD* = 1.30)	1.24 (*SD* = 1.25)
17. Your neighborhood provided the older the ability to enroll in any form of education or training, either formal or non-formal, in any subject in the past year.	1.36*^lowest5^* (*SD* = 1.20)	1.46 (*SD* = 1.47)	0.16*^lowest5^* (*SD* = 0.66)	1.16*^lowest3^* (*SD* = 1.07)	0.88*^lowest1^* (*SD* = 1.17)	1.04*^lowest2^* (*SD* = 1.24)
18. You have access to internet at home.	3.19*^highest1^* (*SD* = 0.97)	1.30*^lowest5^* (*SD* = 1.36)	0.65 (*SD* = 1.12)	0.86*^lowest1^* (*SD* = 1.09)	1.96*^highest4^* (*SD* = 1.61)	1.56 (*SD* = 1.54)
19. You feel safe in your neighborhood.	2.51*^highest3^* (*SD* = 0.79)	3.10*^highest1^* (*SD* = 0.85)	2.96*^highest1^* (*SD* = 1.10)	2.03*^highest3^* (*SD* = 1.04)	2.21*^highest1^* (*SD* = 0.85)	2.61*^highest1^* (*SD* = 1.03)
20. Your neighborhood provided the older participating in an emergency-response training session or drill in the past year which addressed the needs of older residents.	0.40*^lowest1^* (*SD* = 0.89)	0.50*^lowest2^* (*SD* = 0.86)	0.06*^lowest2^* (*SD* = 0.34)	1.18 (*SD* = 1.14)	1.05*^lowest2^* (*SD* = 1.18)	0.58*^lowest1^* (*SD* = 0.97)

x^lowest1^ x^lowest2^ x^lowest3^ x^lowest4^ x^lowest5^ = item that have lowest score of perception as lowest 1 = first lowest, lowest 2 = second lowest, lowest 3 = third lowest, lowest 4 = fourth lowest, and lowest 5 = fifth lowest for each country. X^highest1^ X^highest2^ X^highest3^ X^highest4^ X^highest5^ = item that have highest score of perception as highest 1 = first highest, highest 2 = second highest, highest 3 = third highest, highest4 = fourth highest, and highest 5 = fifth highest for each country.

**Table 3 ijerph-17-04523-t003:** The levels of perceived age-friendly environments by country (applied Pearson chi-squared test analysis).

Item of Perceived Environment	Level of Perception	Country					
		Malay (*N* = 537)	Vietnam (*N* = 497)	Myanmar (*N* = 487)	Thai (*N* = 510)	Japan (*N* = 140)	Total (*N* = 217)
1. The neighborhood is suitable for walking, including for those who use wheelchairs and other mobility aids.	1. Bad (*n*, %)	245	53	303	190	56	847
		45.6%	10.7%	62.2%	37.3%	40.0%	39.0%
	2. Fair (*n*, %)	176	129	110	216	57	688
		32.8%	26.0%	22.6%	42.4%	40.7%	31.7%
	3. Good (*n*, %)	116	315	74	104	27	636
		21.6%	63.4%	15.2%	20.4%	19.3%	29.3%
2. The public spaces and buildings in the community are accessible for all people, including those who have limitations in mobility, vision or hearing.	1. Bad (*n*, %)	273	72	417	214	67	1043
		50.8%	14.5%	85.6%	42.0%	47.9%	48.0%
	2. Fair (*n*, %)	170	164	40	184	61	619
		31.7%	33.0%	8.2%	36.1%	43.6%	28.5%
	3. Good (*n*, %)	94	261	30	112	12	509
		17.5%	52.5%	6.2%	22.0%	8.6%	23.4%
3. The public transport vehicles (e.g., train cars, buses) are physically accessible for all people, including those who have limitations in mobility, vision or hearing.	1. Bad (*n*, %)	330	113	372	264	64	1143
		61.5%	22.7%	76.4%	51.8%	45.7%	52.6%
	2. Fair (*n*, %)	133	159	51	152	59	554
		24.8%	32.0%	10.5%	29.8%	42.1%	25.5%
	3. Good (*n*, %)	74	225	64	94	17	474
		13.8%	45.3%	13.1%	18.4%	12.1%	21.8%
4. The public transportation stops (such as bus stops) are not too far from your home.	1. Bad (*n*, %)	285	102	461	247	44	1139
		53.1%	20.5%	94.7%	48.4%	31.4%	52.5%
	2. Fair (*n*, %)	158	135	8	167	53	521
		29.4%	27.2%	1.6%	32.7%	37.9%	24.0%
	3. Good (*n*, %)	94	260	18	96	43	511
		17.5%	52.3%	3.7%	18.8%	30.7%	23.5%
5. Housing in the neighborhood is affordable.	1. Bad (*n*, %)	233	32	484	120	37	906
		43.4%	6.4%	99.4%	23.5%	26.4%	41.7%
	2. Fair (*n*, %)	267	139	2	283	82	773
		49.7%	28.0%	0.4%	55.5%	58.6%	35.6%
	3. Good (*n*, %)	37	326	1	107	21	492
		6.9%	65.6%	0.2%	21.0%	15.0%	22.7%
6. You feel respected and socially included in your community.	1. Bad (*n*, %)	109	24	331	53	65	582
		20.3%	4.8%	68.0%	10.4%	46.4%	26.8%
	2. Fair (*n*, %)	218	65	35	311	62	691
		40.6%	13.1%	7.2%	61.0%	44.3%	31.8%
	3. Good (*n*, %)	210	408	121	146	13	898
		39.1%	82.1%	24.8%	28.6%	9.3%	41.4%
7. Your neighborhood provided volunteer activity to the older in the last month on at least one occasion.	1. Bad (*n*, %)	213	417	413	207	96	1346
		39.7%	83.9%	84.8%	40.6%	68.6%	62.0%
	2. Fair (*n*, %)	187	22	17	197	21	444
		34.8%	4.4%	3.5%	38.6%	15.0%	20.5%
	3. Good (*n*, %)	137	58	57	106	23	381
		25.5%	11.7%	11.7%	20.8%	16.4%	17.5%
8. You have opportunities for paid employment (i.e., there are opportunities for you to get a paid job if you want for an older person).	1. Bad (*n*, %)	274	302	472	308	104	1460
		51.0%	60.8%	96.9%	60.4%	74.3%	67.3%
	2. Fair (*n*, %)	134	59	8	141	14	356
		25.0%	11.9%	1.6%	27.6%	10.0%	16.4%
	3. Good (*n*, %)	129	136	7	61	22	355
		24.0%	27.4%	1.4%	12.0%	15.7%	16.4%
9. Your neighborhood provided sociocultural activities to the older at least once in the last week.	1. Bad (*n*, %)	240	382	413	143	75	1253
		44.7%	76.9%	84.8%	28.0%	53.6%	57.7%
	2. Fair (*n*, %)	181	30	25	251	37	524
		33.7%	6.0%	5.1%	49.2%	26.4%	24.1%
	3. Good (*n*, %)	116	85	49	116	28	394
		21.6%	17.1%	10.1%	22.7%	20.0%	18.1%
10. You are involved in decision making about important political, economic and social issues in the community.	1. Bad (*n*, %)	414	193	473	259	73	1412
		77.1%	38.8%	97.1%	50.8%	52.1%	65.0%
	2. Fair (*n*, %)	96	117	1	176	40	430
		17.9%	23.5%	0.2%	34.5%	28.6%	19.8%
	3. Good (*n*, %)	27	187	13	75	27	329
		5.0%	37.6%	2.7%	14.7%	19.3%	15.2%
11. Local sources of information about your health concerns and service needs are available.	1. Bad (*n*, %)	158	95	360	143	40	796
		29.4%	19.1%	73.9%	28.0%	28.6%	36.7%
	2. Fair (*n*, %)	173	167	61	200	59	660
		32.2%	33.6%	12.5%	39.2%	42.1%	30.4%
	3. Good (*n*, %)	206	235	66	167	41	715
		38.4%	47.3%	13.6%	32.7%	29.3%	32.9%
12. You have your personal care or assistance needs met in your home setting by government/private care services (i.e., home care nursing/hospice care/non-governmental organization (NGO)/volunteers).	1. Bad (*n*, %)	261	463	471	148	45	1388
		48.6%	93.2%	96.7%	29.0%	32.1%	63.9%
	2. Fair (*n*, %)	208	15	9	186	68	486
		38.7%	3.0%	1.8%	36.5%	48.6%	22.4%
	3. Good (*n*, %)	68	19	7	176	27	297
		12.7%	3.8%	1.4%	34.5%	19.3%	13.7%
13. You have had enough income to meet your basic needs over the previous 12 months without public or private assistance.	1. Bad (*n*, %)	49	141	402	241	67	900
		9.1%	28.4%	82.5%	47.3%	47.9%	41.5%
	2. Fair (*n*, %)	145	97	63	204	50	559
		27.0%	19.5%	12.9%	40.0%	35.7%	25.7%
	3. Good (*n*, %)	343	259	22	65	23	712
		63.9%	52.1%	4.5%	12.7%	16.4%	32.8%
14. Designated priority parking spaces are adequately designed and available.	1. Bad (*n*, %)	253	78	469	324	30	1154
		47.1%	15.7%	96.3%	63.5%	21.4%	53.2%
	2. Fair (*n*, %)	190	141	10	131	61	533
		35.4%	28.4%	2.1%	25.7%	43.6%	24.6%
	3. Good (*n*, %)	94	278	8	55	49	484
		17.5%	55.9%	1.6%	10.8%	35.0%	22.3%
15. Your house has been renovated or can be renovated to fulfil your needs in order to support your activities of daily living.	1. Bad (*n*, %)	167	49	314	273	45	848
		31.1%	9.9%	64.5%	53.5%	32.1%	39.1%
	2. Fair (*n*, %)	202	107	110	156	56	631
		37.6%	21.5%	22.6%	30.6%	40.0%	29.1%
	3. Good (*n*, %)	168	341	63	81	39	692
		31.3%	68.6%	12.9%	15.9%	27.9%	31.9%
16. Your neighborhood provide group physical activities in your leisure time.	1. Bad (*n*, %)	230	300	463	288	61	1342
		42.8%	60.4%	95.1%	56.5%	43.6%	61.8%
	2. Fair (*n*, %)	213	35	11	143	43	445
		39.7%	7.0%	2.3%	28.0%	30.7%	20.5%
	3. Good (*n*, %)	94	162	13	79	36	384
		17.5%	32.6%	2.7%	15.5%	25.7%	17.7%
17. Your neighborhood provided the older the ability to enroll in any form of education or training, either formal or non-formal, in any subject in the past year.	1. Bad (*n*, %)	267	312	467	318	97	1461
		49.7%	62.8%	95.9%	62.4%	69.3%	67.3%
	2. Fair (*n*, %)	191	30	6	143	28	398
		35.6%	6.0%	1.2%	28.0%	20.0%	18.3%
	3. Good (*n*, %)	79	155	14	49	15	312
		14.7%	31.2%	2.9%	9.6%	10.7%	14.4%
18. You have access to internet at home.	1. Bad (*n*, %)	18	320	374	368	58	1138
		3.4%	64.4%	76.8%	72.2%	41.4%	52.4%
	2. Fair (*n*, %)	111	57	55	96	25	344
		20.7%	11.5%	11.3%	18.8%	17.9%	15.8%
	3. Good (*n*, %)	408	120	58	46	57	689
		76.0%	24.1%	11.9%	9.0%	40.7%	31.7%
19. You feel safe in your neighborhood.	1. Bad (*n*, %)	26	25	54	138	21	264
		4.8%	5.0%	11.1%	27.1%	15.0%	12.2%
	2. Fair (*n*, %)	262	57	85	203	67	674
		48.8%	11.5%	17.5%	39.8%	47.9%	31.0%
	3. Good (*n*, %)	249	415	348	169	52	1233
		46.4%	83.5%	71.5%	33.1%	37.1%	56.8%
20. Your neighborhood provided the older participating in an emergency response training session or drill in the past year which addressed the needs of older residents.	1. Bad (*n*, %)	470	459	481	316	89	1815
		87.5%	92.4%	98.8%	62.0%	63.6%	83.6%
	2. Fair (*n*, %)	48	7	4	133	34	226
		8.9%	1.4%	0.8%	26.1%	24.3%	10.4%
	3. Good (*n*, %)	19	31	2	61	17	130
		3.5%	6.2%	0.4%	12.0%	12.1%	6.0%

Note: The results from Pearson chi-squared test show the significant association between country of participants and the level of the perceived age-friendly environment for all 20 items at *p* < 0.05.

**Table 4 ijerph-17-04523-t004:** Predictors of perceived “bad” age-friendly environments, analyzed by ordinal logistic regression with five perceived age-friendly environments as a three ordinal outcome scale (bad, fair, and good) and four predictors (educational level, gender, age level, and country of participants).

Predictors	Perceived Age-Friendly Environments				
	Model 1	Model 2	Model 3	Model 4	Model 5
	Odds Ratio (95% C.I.)	Odds Ratio (95% C.I.)	Odds Ratio (95% C.I.)	Odds Ratio (95% C.I.)	Odds Ratio (95% C.I.)
1. Education					
At least Primary school	0.60 * (0.41–0.88)	0.48 * (0.35–0.65)	0.62 * (0.46–0.83)	0.37 * (0.27–0.50)	0.84 (0.60–1.17)
High school	0.87 (0.60–1.25)	1.08 (0.84–1.40)	0.68 * (0.52–0.90)	0.70 * (0.53–0.93)	1.06 (0.79–1.41)
More than high school (Reference)	1	1	1	1	1
2. Country					
Malaysia	0.24 * (0.15–0.38)	1.53 * (1.00–2.31)	1.27 (0.81–1.98)	0.33 * (0.22–0.49)	0.56 * (0.38–0.81)
Vietnam	0.18 * (0.10–0.31)	1.88 * (1.20–2.93)	1.92 * (1.20–3.08)	3.45 * (2.26–5.25)	0.04 * (0.03–0.70)
Myanmar	0.03 * (0.01–0.08)	0.15 * (0.08–0.28)	0.08 * (0.04–0.16)	0.08 * (0.04–0.16)	0.02 * (0.01–0.00)
Thailand	1.35 (0.84–2.17)	1.61* (1.03–2.53)	1.55 (0.96–2.49)	1.91 * (1.25–2.91)	1.58 * (1.04–2.40)
Japan (Reference)	1	1	1	1	1
3. Gender					
Male	0.94 (0.72–1.21)	0.89 (0.72–1.08)	1.16 (0.95–1.41)	1.34 * (1.10–1.63)	0.78 * (0.63–0.97)
Female (Reference)	1	1	1	1	1
4.Age level					
55–64 years	1.14 (0.79–1.64)	1.63 * (1.20–2.21)	4.97 * (3.51–7.02)	0.96 (0.72–1.28)	0.92 (0.67–1.26)
65–74 years	0.98 (0.69–1.38)	1.22 (0.91–1.64)	2.38 * (1.70–3.34)	0.85 (0.65–1.12)	0.86 (0.63–1.16)
75yrs up (Reference)	1	1	1	1	1

1. * is a significant predictor. 2. *Model 1**, *the dependent variable(DV) is* “participating in an emergency response training session or drill in the past year which addressed the needs of older residents” *Model 2**, *DV is* “enrolling in any form of education or training, either formal or non-formal, in any subject in the past year” *Model 3**, *DV is* “having opportunities for paid employment *Model 4**, *DV is* “involving in decision making about important issues in the community” *Model 5**, *DV is* “having the personal care or assistance needs met in home setting”.

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
