# Peer review of "Age-Friendly Environments in ASEAN Plus Three: Case Studies from Japan, Malaysia, Myanmar, Vietnam, and Thailand"

_ijerph, 2020, doi:10.3390/ijerph17124523_

Round 1

Reviewer 1 Report

This cross-sectional study examined perceived age-friendly environments among older adults in ASEAN plus three countries. The sample included 2,171 adults aged 55+ from these countries. This manuscript addresses an important and interesting topic but it is poorly executed and requires substantial improvement.

The term 'elderly' is pejorative and should not be used. See https://www.bmj.com/content/334/7588/316.abstract for a discussion on this. Please replace all mentions of elderly with an acceptable adjective such as 'older adults'.

The manuscript requires thorough English language proofreading.

Please provide information on the response rate.

How was the survey questionnaire administered? face-to-face, online, mail?

Methods / analytic approach:

It is difficult to make sense of the reasoning behind the analytic choices the authors have made. For example, as every one of the 20 indicators are described separately for each country and then the countries combined, it is terribly difficult to make any sense of this set of results. Furthermore, the main analysis then moves to using a three category outcome variable (queried above) so the very detailed description of the individual indicators adds nothing to the main analysis. My suggestion would be to merge tables 2 and 3 and report them in an appendix or as supplementary material. In the main body, the authors could report the % in each country, and overall, in each of the bad, fair, good groups. That is, assuming a justification for these categories (queried above) is provided.

How exactly were the categories of bad, fair and good age-friendliness decided? Are these based on tertiles or something else? Also, why did the authors chose these categories instead of organising their analysis around the established domains of the age-friendly measure?

The choice of indicators is very unclear (Table 4). The authors say that "Five environments (from 20 items) from 3.3 were selected to analyze based on the score with lowest", but I don't know what this means. Why 5 indicators? why these 5? what does 3.3 refer to?

The main statistical method was an ordinal logistic regression (Table 5). However, the sampling approach (multi-stage, stratified sampling) requires that a multilevel modelling approach is used. This is because the observations are not independent but rather clustered within the countries sampled. The multilevel analysis needs to account for four levels: country, city, district, individuals.

Table 4. I am not entirely clear as to what this analysis is trying to say and therefore can not know for sure is Pearson's s the correct approach. regardless, the small cell sizes reported mean that Pearson's chi-square is most likely invalid.

Table 5. Please report the coefficients for all estimates so that the reader can assess the effect size. The p values alone tell us little of value.

To see a clearer analytic approach in a similar study, see https://doi.org/10.1080/13607863.2016.1154011

Due to the difficulties described above with assessing the results, it is not possible to assess the veracity of the conclusions the authors draw from their analysis. Apart from this, there are some other issues with the discussion section that should be addressed. The key concept of social capital is introduced here for the first time (Line 276). This should have been introduced in the background section. Similar for respect. There are other problems with the discussion similar to these but again, they are difficult to comment on in light of the unclear analysis.

Author Response

Reviewer 1 (In the manuscript, change was made with the blue sentences)

  1. The term 'elderly' is pejorative and should not be used. See https://www.bmj.com/content/334/7588/316.abstract for a discussion on this. Please replace all mentions of elderly with an acceptable adjective such as 'older adults'.

Replace all term “elderly”

  1. The manuscript requires thorough English language proofreading.

Already checked by a native academic writer

  1. Please provide information on the response rate.

Response rate 95% (it was added under 3.1 sample characteristics).

  1. How was the survey questionnaire administered? face-to-face, online, mail?

This research design was a cross sectional interview household survey of perceived age-friendly environment towards the older population in ASEAN plus three. The study used a mustistage, stratified sampling procedure collecting data via face to face interview during November 2018 to January 2019 in five metropolitans of Malaysia, Myanmar, Vietnam, Thailand, and Japan.

  1. Methods / analytic approach:

It is difficult to make sense of the reasoning behind the analytic choices the authors have made. For example, as every one of the 20 indicators are described separately for each country and then the countries combined, it is terribly difficult to make any sense of this set of results. Furthermore, the main analysis then moves to using a three category outcome variable (queried above) so the very detailed description of the individual indicators adds nothing to the main analysis. My suggestion would be to merge tables 2 and 3 and report them in an appendix or as supplementary material. In the main body, the authors could report the % in each country, and overall, in each of the bad, fair, good groups. That is, assuming a justification for these categories (queried above) is provided. 

 We already merge tables 2 and 3 and rewrite the results for easily understanding as the recommendation.

How exactly were the categories of bad, fair and good age-friendliness decided? Are these based on tertiles or something else? Also, why did the authors chose these categories instead of organising their analysis around the established domains of the age-friendly measure?

 In this study, each individual item of perceived age-friendly environments is scored from 0 to 4 on a response ordinal scale (not at all, a little, moderately, mostly, extremely), with higher scores indicating a higher perceived of age-friendly environments. In this study, to make it concisely interpretation, we classified the perceived age-friendly environments from five level (not at all, a little, moderately, mostly, extremely), into three level as bad (not at all/a little), fair (moderately), and good (mostly/extremely).

The choice of indicators is very unclear (Table 4). The authors say that "Five environments (from 20 items) from 3.3 were selected to analyze based on the score with lowest", but I don't know what this means. Why 5 indicators? why these 5? what does 3.3 refer to?

For table 4 (now it was table 3 because we merge table 2 and 3 as table 2), we select five items based on the value of average score from first lowest to five lowest. To make it easily for understanding, we already add them all 20 items, instead of selecting five items which get the five lowest score.

The main statistical method was an ordinal logistic regression (Table 5). However, the sampling approach (multi-stage, stratified sampling) requires that a multilevel modelling approach is used. This is because the observations are not independent but rather clustered within the countries sampled. The multilevel analysis needs to account for four levels: country, city, district, individuals.

In this study, a study’s population of interest is massive, the standard sampling procedure, random sampling, becomes infeasible. Therefore, we select to use multi-stage sampling to avoid the problems of randomly sampling from a population that is larger than our resources can handle. This sampling procedure in essence is a way to reduce the population by cutting it up into smaller groups, which then can be the subject of random sampling. We assume that our sample population have low between-group variance, therefore this form of sampling should be a legitimate way to simplify the population. We selected to analyze the data via ordinal logistic regression because our research question is to investigate high risk groups of having inadequate age-friendly environments that all are individual level.  

Table 4. I am not entirely clear as to what this analysis is trying to say and therefore cannot know for sure is Pearson's s the correct approach. regardless, the small cell sizes reported mean that Pearson's chi-square is most likely invalid. 

The results of Pearson's chi-square as Table 4 (a current table3) are valid for all 20 items.

 Of 18 items from 20 items, 0 cells (0.0%) have expected count less than 5.

Other 2 items, 1 cells (4.0%) have expected count less than 5.

Table 5. Please report the coefficients for all estimates so that the reader can assess the effect size. The p values alone tell us little of value.

Table 5(current table 4), the coefficients for all estimates are added as the recommendation.

  1. Due to the difficulties described above with assessing the results, it is not possible to assess the veracity of the conclusions the authors draw from their analysis. Apart from this, there are some other issues with the discussion section that should be addressed. The key concept of social capital is introduced here for the first time (Line 276). This should have been introduced in the background section. Similar for respect. There are other problems with the discussion similar to these but again, they are difficult to comment on in light of the unclear analysis.

The discussion was revised to make it clearer and smoother comparing to the previous one, I hope that a reader can access easier.

Reviewer 2 Report

The contribution presents interesting research about the perceived age-friendly environments through a quantitative analysis of surveys spread in Malaysia, Vietnam, Myanmar, Thailand, and Japan that involved 2,171 older adults. Elaborated data provide a set of insights that can support the policy-making process at different levels.

Despite the interest of the topic and the proposed approach, I would propose to the authors a set of suggestion in order to improve the quality of the article.

The introduction lacks a clear definition of the final target of the study. Over the presentation of the theoretical framework, I think the authors should point out how these studies could contribute to the field of research and provide useful knowledge for the policy-community. In addition, the introduction should present the overall structure of the article.

Section 2. Together with the description of survey and study population, measures and data analysis, the authors could point out in this section the reasons of the approach applied for the panel of questions.  

Section 3. provides the results' presentation. They are precise and interesting outcomes from the survey, but in some point the description became didascalic and may it could be integrated with a previous discussion of specific aspects that emerge from each table.

Conclusions could go deeper in the discussion of the results of the study, introducing a critical reading of the methodology and the use of a quantitative approach in such a complex context.

Author Response

Reviewer 2(In the manuscript, change was made with the red sentences)

  1. The introduction lacks a clear definition of the final target of the study. Over the presentation of the theoretical framework, I think the authors should point out how these studies could contribute to the field of research and provide useful knowledge for the policy-community. In addition, the introduction should present the overall structure of the article.

We add the clear definition of the final target of the study. We already pointed out how these studies could contribute to the field of research and provide useful knowledge for the policy-community. In addition, we present the overall structure of the article in the introduction part.

  1. Section 2. Together with the description of survey and study population, measures and data analysis, the authors could point out in this section the reasons of the approach applied for the panel of questions.  

For methodology, we give the reasons of the approach applied for the panel of questions. 

  1. Section 3. provides the results' presentation. They are precise and interesting outcomes from the survey, but in some point the description became didascalic and may it could be integrated with a previous discussion of specific aspects that emerge from each table.

In order to provide the results clearer, we merge table 2 and 3, and rewrite results more interesting.

  1. Conclusions could go deeper in the discussion of the results of the study, introducing a critical reading of the methodology and the use of a quantitative approach in such a complex context.

 We add the detail in the discussion of the results of the study focusing a quantitative approach in the first paragraph of the discussion part.

Round 2

Reviewer 1 Report

I wish to thank the authors for the amendments they have made to their original submission. Some minor remaining issues are.....

The term elderly is still used in section 3.4, page 15 "the results indicated that elderly with more than high school are". I can not state strongly enough how problematic this is. Please remove it.

The authors have misinterpreted my comment regarding presenting all coefficients in the original Table 5 (new Table 4). Maybe my comment was unclear, but what I suggested was that the Odds Ratios be reported for all results, significant and non-significant so that effect sizes can be compared.

The sampling procedure is still a little unclear. The multi-stage method is clear enough but there is no explanation as to why the sample size for Japan is so much smaller than the others (n=140). Furthermore, the 95% response rate is phenomenal. Was this the same in each area?

Author Response

Dear Reviewer,

Please kindly consider our response as follows:

1.The term elderly is still used in section 3.4, page 15 "the results indicated that elderly with more than high school are". I can not state strongly enough how problematic this is. Please remove it.

Yes, we already changed the term.

2.The authors have misinterpreted my comment regarding presenting all coefficients in the original Table 5 (new Table 4). Maybe my comment was unclear, but what I suggested was that the Odds Ratios be reported for all results, significant and non-significant so that effect sizes can be compared.

Yes, we already adjusted the results as the recommendation.

3.The sampling procedure is still a little unclear. The multi-stage method is clear enough but there is no explanation as to why the sample size for Japan is so much smaller than the others (n=140). Furthermore, the 95% response rate is phenomenal. Was this the same in each area?

We add the detail of Japan's data collection as this.... After adding a missing rate of 25%, the expected final sample size for each metropolitan area was approximately 500 cases. However, we decided to have smaller numbers of data collection (about 150 cases) in Japan due to our resource limitation.

We also added The response rate for each country was 100% . The study population after excluding the observations with missing data was a total of 2,171 persons aged 55 years and older.  (No one did reject our face-to-face interview questionaires, but we still have some missing questions that they did not answer.)

Thank you very much for your kindly recommendations,

Warmest Regards,

Authors

Reviewer 2 Report

I think the authors improved all the weaker points that I mentioned in the previous review and now the manuscript is publishable

Author Response

Dear Reviewer,

Thank you very much for your recommendations. We are highly appreciated.

Warmest Regards,

Authors